# Electrochemical Response of 3D-Printed Free-Standing Reduced Graphene Oxide Electrode for Sodium Ion Batteries Using a Three-Electrode Glass Cell

**DOI:** 10.3390/ma16155386

**Published:** 2023-07-31

**Authors:** Cristina Ramírez, María Isabel Osendi, Juan José Moyano, Jadra Mosa, Mario Aparicio

**Affiliations:** 1Institute of Ceramics and Glass (ICV), Consejo Superior de Investigaciones Científicas, CSIC, Kelsen 5, Cantoblanco, 28049 Madrid, Spain; miosendi@icv.csic.es (M.I.O.); moyanosubires.jj@gmail.com (J.J.M.); jmosa@icv.csic.es (J.M.); maparicio@icv.csic.es (M.A.); 2Universidad Politécnica de Madrid, 28040 Madrid, Spain

**Keywords:** reduced graphene oxide (rGO), Na ion battery, rGO electrode, direct ink writing

## Abstract

Graphene and its derivatives have been widely used to develop novel materials with applications in energy storage. Among them, reduced graphene oxide has shown great potential for more efficient storage of Na ions and is a current target in the design of electrodes for environmentally friendly Na ion batteries. The search for more sustainable and versatile manufacturing processes also motivates research into additive manufacturing electrodes. Here, the electrochemical responses of porous 3D-printed free-standing log-type structures fabricated using direct ink writing (DIW) with a graphene oxide (GO) gel ink are investigated after thermal reduction in a three-electrode cell configuration. The structures delivered capacities in the range of 50–80 mAh g^−1^ and showed high stability for more than 100 cycles. The reaction with the electrolyte/solvent system, which caused an initial capacity drop, was evidenced by the nucleation of various Na carbonates and Na_2_O. The incorporation of Na into the filaments of the structure was verified with transmission electron microscopy and Raman spectroscopy. This work is a proof of concept that structured reduced GO electrodes for Na ion batteries can be achieved from a simple, aqueous GO ink through DIW and that there is scope for improving their performance and capacity.

## 1. Introduction

Carbon-based materials have influenced the development of electrodes for energy conversion and storage devices, with graphite being the primary choice for anode production due to its natural abundance, low cost, high electrical conductivity, and moderate capacity [1]. However, the increasing availability of graphene is shifting the balance towards the use of this material and its derivatives in a new generation of energy applications due to its high stiffness, high surface area, and facile functionalization. In addition, these properties make graphene sheets ideal to form hybrids with metal nanoparticles and metal oxide compounds, achieving enhanced electrochemical performance and better control of electrode swelling [2,3,4].

Currently, the greatest efforts to develop advanced energy materials are made in the context of rechargeable batteries, as lithium ion batteries (LIBs) represent more than 85% of energy storage devices [5]. This is being achieved through two different approaches: the search for LIB components that can deliver higher energy density and capacity and the replacement of Li itself with an environmentally friendly alternative, such as the sodium ion battery (SIB). Research into the latter has increased sixteen-fold in the last ten years (according to WOS) and is motivated by a number of advantages, such as the reduced production costs due to the abundance of sodium, the replacement of Cu current collectors with Al, the high energy density [6], and the chemical stability of more substances in direct contact with Na metal [7].

The potential of graphene materials in the fabrication of novel Li ion battery components has been demonstrated in several works [8,9,10,11], showing their suitability for improving ion diffusion, electron transport, stability, and rate performance, as well as highlighting some essential issues still under research, such as the high irreversible capacity caused by the formation of the solid electrolyte interphase (SEI).

For SIBs, the use of graphite and pristine graphene electrodes does not seem to produce beneficial effects on the Na^+^ intercalation process due to the instability of Na-graphite intercalation compounds [5], and studies to modify the micro- and nanostructure of carbonaceous electrodes have recently intensified [12,13]. Indeed, it has been observed that pore size and morphology, as well as the defective/functionalized surface of reduced graphene oxide (rGO), create appropriate conditions for improving electrolyte wetting and Na adsorption–insertion [14,15]. For instance, Luo et al. [16] compared the performances of rGO, CNT, graphitic microbeads, and activated carbon, finding the best response for the rGO-based electrode, achieving 220 mAh g^−1^ of discharge capacity at 30mA g^−1^, and retaining 80% of the initial capacity after 300 cycles. N-doped rGO foam anodes [17] also showed better performances than N-doped graphene and pure rGO, delivering an initial reversible capacity of 853 mAh g^−1^ at 500 mA g^−1^, with 70% of the initial charge capacity after 150 cycles. The reason for this behavior was attributed to the presence of oxygen functional groups that interacted with Na^+^, favoring the insertion/extraction capability. Ali et al. fabricated an rGO cathode as an alternative to electrodes based on transition metals [18]. At 30 mA g^−1^, the cathode showed a low coulombic efficiency attributed to the oxidative degradation of the electrolyte, but it showed stability for more than 1000 cycles, maintaining a discharge capacity of more than 235 mAh g^−1^, and it also exhibited 134 mAh g^−1^ at 600 mA g^−1^.

Battery fabrication technology was recently reviewed by Lyu et al. [19], who showed that additive manufacturing (AM) can provide significant contributions to the challenges of future batteries: (i) a reduction in production cost by eliminating some machining steps and reducing waste, and (ii) the fabrication of complex architectures required by miniature electronic devices and novel applications while achieving high energy density by maximizing the surface-to-volume ratio. When comparing different 3D-printing methods, direct ink writing (DIW) stands out as the most versatile, covering different types of materials and composites, designs, and sizes and providing an adequate resolution [19]. In the case of SIBs, AM could also help overcome the challenge of designing electrodes with specific porosity that allow better access to sodium ions, as has been demonstrated with rGO foams [14,17], which can be exploited in the fabrication of flexible designs intended for energy-harvesting applications [20]. Nevertheless, the 3D printing of graphene electrodes for SIBs is still an under-explored area, and only a few papers can be cited. Yu and coworkers [21] developed an rGO current collector with patterned macroholes and ice-templated microporosity to be used with a Na metal anode. The printed structure improved the stability of the Na plating/stripping process, which was controlled by the hole regions, achieving capacities of 627 mAh g^−1^. More recently, Yang et al. [22] fabricated N-doped 3D-printed rGO microlattices to be assembled in a Na metal anode, which inhibited dendrite formation and achieved a stable profile for 500 h at a current density of 5 mA cm^−2^ with a specific capacity of 748 mAh g^−1^. In the same line, Yan and coworkers [23] developed 3D Na-rGO/CNT hybrid anodes, which allowed a more uniform distribution of current density and effectively accommodated volume changes during cycling. The cell provided a discharge capacity of 93 mAh g^−1^ at 100 mA g^−1^ at the seventh cycle. Three-dimensional printed structures were also used to design rGO cathodes for Na air batteries with channels and pores that allowed effective O_2_ transport, yielding a high capacity of 1.26 mAhcm^−1^ at 0.2 C with a Na_3_V_2_(PO_4_)_3_/rGO composite structure [24], as well as a discharge capacity of 500 mAh g^−1^ at 1 A g^−1^ that was stable for 120 cycles with pure rGO [25]. Wang et al. also improved the performance of a Na battery by the attachmentof a 3D-printed rGO structure decorated with Au nanoparticles to the Na metal anode, preventing dendrite formation [26].

The properties of printed GO electrodes reported above have been measured with thin graphene frameworks tested with different electrolytes using coin cell or Swagelok configurations. These experimental setups are not appropriated for studying the responses of 3D thick, multilayer structures, which are relevant to the design of future energy storage systems that require interdigitated, coaxial, lattice, and non-planar configurations, which can also be infiltrated by polymer, liquid, and gel electrolytes [27].

In this work, we choose to use large, pure rGO scaffolds printed using DIW with rGO inks prepared from commercially available GO powders. The 3D rGO log-type lattices were tested as self-supported electrodes without a metallic current collector (working electrode) using Na as the counter and reference electrodes in a three-electrode glass cell, which allows the measurement of electrochemical performance without applying any compressive stresses to the structure. The aim is to gain new insights into reversible sodium ion insertion into thick rGO scaffolds. A complete characterization of the rGO electrodes after various charge/discharge cycles is performed to reveal possible degradation mechanisms affecting cell performance, as well as whether this degradation is similar to that observed in hard carbon electrodes, to which rGO is an alternative material.

## 2. Materials and Methods

### 2.1. Three-Dimensional Printing of rGO Scaffold

The ink preparation and subsequent printing processes were described in detail in a previous paper [28]. In brief, a hydrogel-type ink was prepared by thoroughly mixing 5.1 wt. % GO nanosheets (GO, grade N002-PDE, Angstron Materials Inc., Dayton, OH, USA), 1.9 wt. % poloxamer triblock copolymer (Poloxamer 407, Sigma-Aldrich, Merck, Darmstadt, Germany), and 93.0 wt. % ultrapure water (pH was adjusted to 5.2 with a HCl droplet) in a planetary centrifugal mixer using nylon balls (Figure 1a). The ink was loaded into a printing syringe and then extruded though a nozzle with a 410 µm diameter (Precision Tips; EFD Inc., Westlake, OH, USA) with a three-axis robocasting system (A3200, 3-D Inks LLC, Tulsa, OK, USA) using the design software of the equipment. Lattices with external dimensions of 13.5 × 13.5 × 6.5 (all in mm) were printed by depositing a linear array of parallel filaments layer by layer until 16 layers were completed on a flat alumina substrate that allowed easy removal (Figure 1b). The printed cells were left to dry in ambient air for 24 h and then treated at 1200 °C (2 h) in a graphite furnace (Astro, Thermal Technology Inc., Santa Rosa, CA, USA) under a N_2_ atmosphere for poloxamer removal and GO reduction purposes, named as reduced GO (rGO), after this treatment [28]. The structures mostly shrunk during drying, and their final external dimensions were 7.0 × 7.0 × 2.6 (in mm) with a logpile structure (Figure 1c).

### 2.2. Electrochemical Measurements

Electrochemical characterization of the 3D-printed rGO electrode (attached to a Pt wire at a single point using conducting silver paint) was performed in an argon-filled glass cell using Na foils as counter and reference electrodes (Figure 1d). An amount of 1 M of NaClO_4_ solution in propylene carbonate (PC) was used as electrolyte. Cyclic voltammetry (CV) was performed between 0.010 and 2.000 V vs. Na^+^/Na at 6.0 mV min^−1^. Galvanostatic charge–discharge cycles were studied between the same potentials at different current intensities. Both tests were performed using a Bio-Logic VMP3 multichannel potentiostat. Electrochemical impedance spectroscopy (EIS) tests were performed in the frequency range from 1 MHz to 100 MHz using an AC amplitude of 5 mV before and after charge–discharge cycling at 10 mA g^−1^.

### 2.3. Microstructural Characterization

The 3D samples after electrochemical tests and, in particular, after a charging step and after a discharging step were characterized using different techniques. They were analyzed with X-ray diffraction (XRD) procedures (D8 Advance diffractometer, Bruker Corp -Karlsruhe, Germany) on the top layers of the 3D-printed structures and on previously powdered parts. Microstructure observations of the 3D samples were performed with a field emission scanning electron microscope (FESEM, S-4700, Hitachi, Tokyo, Japan) equipped with energy-dispersive X-ray spectroscopy (EDS) for microanalysis. Observations of the rGO electrodes with a high-resolution transmission electron microscope (HRTEM, JEOL 2100, Akishima shi, Japan) were completed after gentle comminution, ultrasonic dispersing in isopropanol, and drop-casting of the particle dispersion on a Cu grid. Selected area diffraction patterns (SADPs) were also taken from representative areas. Micro-Raman analyses (Alpha 300-R, WITec, Ulm, Germany) of polished and unpolished samples were performed on the surfaces of the 3D electrode filaments using two scan modes with an acquisition time of 60 ms per spectrum, a surface scan for areas of 15 × 15 µm^2^ to obtain average spectra, and a depth scan along a line and 3 µm in depth from the surface. Separately, Raman spectra were also acquired along filament cross-sections using line scans.

## 3. Results and Discussion

The printed samples were lightweight 3D structures with a geometric density, ρ_g_, of 0.13 g cm^−3^ and a total porosity of ~95% (Table 1), which accounted for the large voids of the lattice design (Figure 2a) and the porosity associated with the struts [28]. The macroporous design with channels of about 250 μm and the high porosity within the struts (about 83%) were intended to facilitate the electrolyte access to the entangled multilayer rGO electrode (Figure 2a, Table 1). The filaments had a diameter of about 260 μm and were formed from intertwined rGO sheets with lateral dimensions below 7 µm (Figure 2b,c). The thermal treatment at 1200 °C was chosen because it has previously been shown to be effective in reducing the initial GO oxygen level (~1 wt. % after thermal treatment) and increasing the electrical conductivity [28,29].

The cyclic voltammetry of a cell with a printed rGO electrode is plotted in Figure 3 for seven consecutive cycles. Cycle 1 presented two significant cathodic peaks centered at 0.2 and 1.0 V, mainly associated with different processes related to the reaction between the electrolyte and the rGO electrode, generating the formation of the SEI [16,17,30,31]. The following cycles showed significant reductions in these peaks, indicating the progressive stabilization of the SEI film. Instead, a broad cathodic peak around 0.5 V was observed, associated with the insertion of sodium ions into the graphene structure in combination with the oxidation of the sodium foil (discharge process). This peak decreased with the number of cycles until it stabilized, possibly indicating that equilibrium between the sodium ion insertion process and the SEI formation was reached. The oxidation part of the curves corresponded to the charging process with the de-insertion of sodium ions and their reduction on the sodium foil. An anodic peak was observed centered at 0.5 V, which increased with the number of cycles until it stabilized according to a similar process observed during the discharge.

Galvanostatic charge–discharge curves at different current densities between 0.010 and 2.000 V (vs. Na^+^/Na) are shown in Figure 4. The results in Figure 4a display a pronounced reduction in the charge–discharge capacities for all the current intensities. However, a high stability of the capacity values was observed from cycle 36 until the end of the test. This behavior may be due to the fact that a high number of cycles was needed to stabilize the electrochemical system, including the formation of the SEI. The achieved capacity values are not very high compared to reported data focused on graphene materials with N doping or containing molecular spacers [16,17,30,31], but it must be emphasized that, in the present case, 3D free-standing structures of 16 layers with relatively thick filaments (250 μm) were evaluated. An explanation is that not all of the 16 layers contributed effectively to the capacity due to an occasional lack of connectivity during the electrochemical tests, as the postmortem SEM images (below) show [20]. Moreover, the nodes where the filaments crossed were zones of potential ohmic losses, as demonstrated for other conducting scaffolds [32]. Indeed, the 3D structure can be adapted in future designs to address these problems [27].

It should also be noted that no other compounds or dopants were used to improve the electrical conductivity of these 3D rGO samples, which was ~800 S m^−1^ for the present structures, as previously reported [28]. Nevertheless, the present hydrogel ink could be filled with other particles (dopants, spacers), taking advantage of the triblock copolymer characteristics; in particular, mixtures of GO and pristine graphene nanoplatelets were effectively printed [28], modifying the scaffold shrinkage behaviour and electrical conductivity. Accordingly, the present research should be considered as an enrichment in the data demonstrating the feasibility of DIW for the fabrication of structured graphene electrodes from hydrogel inks for Na ion batteries. Last but not least, a possible opposite effect of the GO reduction temperature on the insertion process may occur by restacking the rGO sheets, as pointed out by other authors [33]. Furthermore, the standalone nature of the present electrode and the absence of any other elements or additives facilitated its plain characterization after the electrochemical tests.

Figure 4b presents the smooth charge and discharge profiles for different current intensities corresponding to the insertion and extraction of sodium ions in the graphene structure. These profiles clearly correspond to the broad peaks observed in the oxidation and reduction of CV at around 0.5 V in both cases. The results of Figure 5 show a greater stability of the capacity, with decreases from 76 to 53 mAh g^−1^ after more than 150 charge–discharge cycles. During the first cycles, a larger drop in capacity was observed due to the formation of the SEI, but after approximately 30 cycles, the capacity values were quite stable. On the other hand, the low current density used favored the formation of a stable SEI and subsequent, more homogeneous cycling.

Figure 6 shows the impedance spectra (Nyquist plots) before and after discharge–charge cycling at 10 mA g^−1^. The spectra show an inductive component at high frequencies, followed by a resistive component, two semicircles, and a diagonal line regarding typical diffusion behavior at lower frequencies. The inductor was related to the wiring between the electrodes with the equipment, and it was not taken into account in the adjustment of the impedance spectra. Consequently, the equivalent circuit used is shown in Figure 6b. R1 represents the resistance associated with the liquid electrolyte and could be estimated by the intersection between the Nyquist curve and the x-axis. Each electrode–electrolyte interface had a double-layer capacitance (Q2, Q3) and a charge transfer resistance (R2, R3). Finally, the Warburg element (W4) was related to the Na ion diffusion inside the 3D rGO self-supported electrode. The figure shows the fit of both spectra as a solid line. The fitting process indicated a small increase in the R1 value (from 33 to 36 Ohm) due to conductivity reduction because of the gradual degradation of the electrolyte. The increments in the values of R2 and R3 were a little higher, going from 2.2 to 6.1 Ohm and from 16 to 20 Ohm, respectively, and were related to the usual aging of electrodes after the cycling process. However, from the results, it can be noted that the deterioration of the 3D rGO electrode was very small considering the high number of charge–discharge cycles performed.

The XRD patterns of the discharged and charged 3D rGO electrodes showed a broad amorphous band at ~25.6° (2θ) attributable to the (002) graphitic peak. In addition, sodium carbonate byproducts were found in both the charged and discharged samples. The quality of the spectra, with low intensity and poor crystallinity, allowed only the identification of these compounds, with a certain prevalence of the nahcolite phase (see Appendix A). The presence of this phase has been reported as a discharge product in rechargeable Na–CO_2_/O_2_ batteries [34,35], while the carbonate (Na_2_CO_3_) has been described as the main product in polycarbonate (PC)-containing electrolytes, resulting from the decomposition of this solvent [36,37]. The present data confirm the occurrence of some reaction with the PC solvent, which was previously addressed in hard carbon anodes through passivation with fluorinated compounds in the electrolyte solution [38]. This decomposition of a PC-NaClO_4_ electrolyte system on the surface of hard carbon electrodes was suggested to be responsible for a blocking effect on the surface and an increase in the internal resistance of the SEI [39], thus affecting the final capacity.

Figure 7a,b show SEM images of the rGO electrodes, revealing that, although the structure grid pattern was preserved, some filaments appeared broken after electrochemical testing and manipulation. Some sodium-containing precipitates coming from the electrolyte appeared unevenly on the filament surface of the electrode (see SEM images in Appendix A).

EDS microanalyses of the filament diameter cross-sections indicated the presence of Na inside the filaments for both the charged and discharged electrodes (Appendix A), confirming that the wire thickness would not hinder the penetration of the electrolyte through the porous 3D filament, as this penetration was affected by restacking of the graphene layers after thermal reduction. An EDS microanalysis performed on the polished electrodes after embedding them in epoxy resin to obtain flat cross-sections also revealed the penetration of Na ions inside the filaments of the 3D electrodes, although the requirement of a thin Au coating biased the measurement (see Appendix A). According to the Na line profile microanalysis of uncoated samples, it can be estimated that the Na concentration fluctuated around 8 wt. % (±4) within the filaments of both 3D rGO electrodes (Appendix A). A significant concentration of Na atoms was also detected on the surface of the discharged electrode (Appendix A), which decreased toward the inner filament to the levels indicated above compared to the concentration of C atoms throughout the filament. This high concentration of sodium ions on the surface of the material may also be due to the sodium salt of the electrolyte deposited on the surface after evaporation of the solvent, as the eventual presence of Cl in some areas evidenced.

EDS elemental maps (C, Na, O) for the charged 3D electrode show rather uniform concentrations of Na and O in the first 40 µm from the rod fracture surface (Appendix A), while Na seemed to be concentrated mainly on the surface of the discharged electrode (see Appendix A). Some elements, such as Cl and K, also seemed to be concentrated near the filament surface region (~15 μm from the rim). Both elements were contaminants present in the original samples (see Appendix A) coming from the graphite oxidation for GO production, and Cl may also be associated with the electrolyte, as mentioned before.

A general TEM view of the discharged 3D rGO electrode is displayed in Figure 8a, where the graphene sheets appear intertwined, showing the typical SAED ring pattern of graphene. A few darker areas between the sheets showed small crystals that also aggregated into larger areas (Figure 8b). These crystals of ~20 nm showed a cubic habit and were identified according to the SAED as Na_2_O, probably from the decomposition of the electrolyte solvent and its reaction with sodium ions. Thanks to the fact that the cell was transparent, it was possible to observe places where the electrodes were submerged, which turned yellow, indicating the degradation of the solvent [40]. These crystals were not observed in the charged sample (see Appendix A), which may indicate a preferential nucleation of this reaction product in certain areas but no absence. In situ TEM observations by Wan et al. of the charge–discharge cycle in a GO/Na cell also demonstrated the formation of Na_2_O, described as an irreversible phase [41]. Sodium oxides were reported as common reaction products for Na air batteries, with the peroxide Na_2_O_2_ and super peroxide NaO_2_ being the most commonly described in Na air batteries [7].

HRTEM images of stacked graphene sheets in both charged and discharged rGO electrodes indicated a range of graphene interplanar spacings (*d*), with the charged rGO electrode having *d* in the range of 0.42–0.47 nm while the discharged electrode had a typical *d* in the range of 0.42–0.43 nm (Figure 9). Both were larger than the ionic radius of Na (0.098 nm), thus allowing Na insertion. It is important to note that, for both samples, *d* was higher on average than the measured interplanar distance of untested rGO at ~0.38 nm, thus evidencing Na insertion between the rGO sheets.

The average Raman spectra of the depth scan along the filament surface of the 3D rGO electrodes penetrating ~3 μm below the surface are plotted in Figure 10a. The usual bands of graphene-based materials (D, G, 2D) were observed in the spectra, with a large fluorescence band in the case of the discharged sample. Both samples had a relatively large D band associated with defects in the rGO. In Table 2, the I_D_/I_G_ ratio obtained through peak integration appeared for both the charged (1.9) and discharged (1.7) samples, having relatively close ratios, while the G band’s position of the discharged sample was redshifted with respect to the charged one. For a better evaluation of this effect, Figure 10b shows the Raman shift maps, revealing the statistical redshift of the G band to lower frequencies in the discharged electrode, a result that is in agreement with data reported by Reddy et al. [42] through in situ measurements in hard carbon samples intercalated with Na. A similar behavior was observed for Na and Li insertion in hard carbon, which was associated with the elongation of C–C bonding by occupying the π* antibonding orbital [43]. This effect is also confirmed by Raman radial scans obtained for the polished cross-sections of different filaments (Figure 10c,d) in the two electrodes, which showed differences in the G band positions of 5–10 cm^−1^. Table 3 summarizes the magnitude of the redshifts reported in different works, which are associated with the type of carbon used and the stage of insertion achieved (SEI formation, intercalation, or pore occupancy) [44]. These results provide indirect evidence for the insertion/de-insertion of Na^+^ between the rGO layers.

The Raman spectra recorded directly from the unpolished filament surfaces of the charged and discharged 3D rGO samples showed similar shifts. When compared with the Raman spectra of the original 3D rGO sample (see Appendix A), a certain redshift of the D and G bands was detected for both the charged and discharged samples, which could be attributed to strain or doping effects [45,46] associated with Na insertion/de-insertion.

Table 4 presents a comparison of the capacities and cycle lives achieved for different configurations of rGO electrodes. It is a fact that rGO is an interesting option in the fabrication of carbon-based electrodes for SIBs, but it is important to note that conventionally fabricated electrodes include additives that improve the electrical conductivity and integrity of the bulk and that foam/3D rGO-based structures commonly reported are used as hybrids with different active materials. This study serves as a proof-of-concept of the electrochemical responses of thick, pure 3D rGO electrodes from which the optimization of designs, reduction treatments, or alternative electrolyte systems [48] should be pursued.

## 4. Conclusions

3D lattices of rGO were obtained using DIW with a hydrogel ink and were used as bare electrodes in a Na ion cell. Although rather moderate capacities were obtained (80 mAh g^−1^), the results showed the insertion/de-insertion of Na ions in the rods of the self-supported 3D graphene networks and good stability under cycling (150 cycles). Electrolyte stability issues, which have been observed in similar materials such as hard carbons, should be addressed by exploring alternative solvents/electrolytes. The DIW of commercial GO material was probed as a straightforward method to achieve electrodes for Na batteries, but the synthesis method should be improved to increase the capacity of the cell by introducing spacers, other graphene compounds, or metal oxides, as well as more sensible designs that avoid/reduce filament crossing and enhance microporosity. 

## Figures and Tables

**Figure 1 materials-16-05386-f001:**
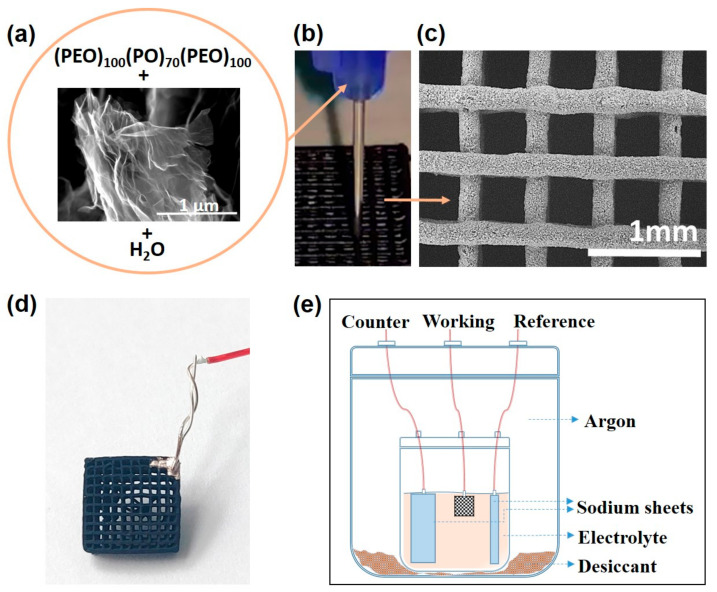
Steps of the process followed to fabricate and characterize the 3D rGO self-supported electrodes. (**a**) Ink components: GO, poloxamer (PEO: polyethylene oxide; PO: polypropylene oxide), and water. (**b**) Printing by robocasting; (**c**) top view of the characteristic grid of the 3D rGO sample; (**d**) electrode connected to Pt wire; and (**e**) schematic figure of the three-electrode glass cell for electrochemical characterization.

**Figure 2 materials-16-05386-f002:**
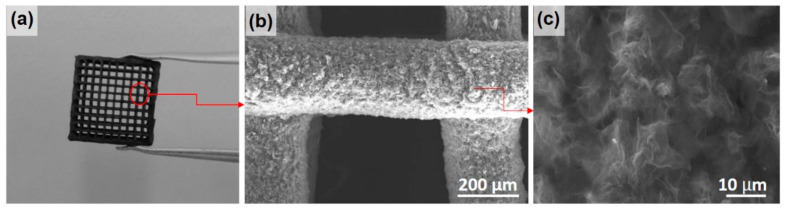
Representative images of the 3D rGO samples used in electrochemical tests showing (**a**) grid pattern, (**b**) characteristic filament diameter, and separation; (**c**) a higher-magnification image of filaments with entangled rGO sheets.

**Figure 3 materials-16-05386-f003:**
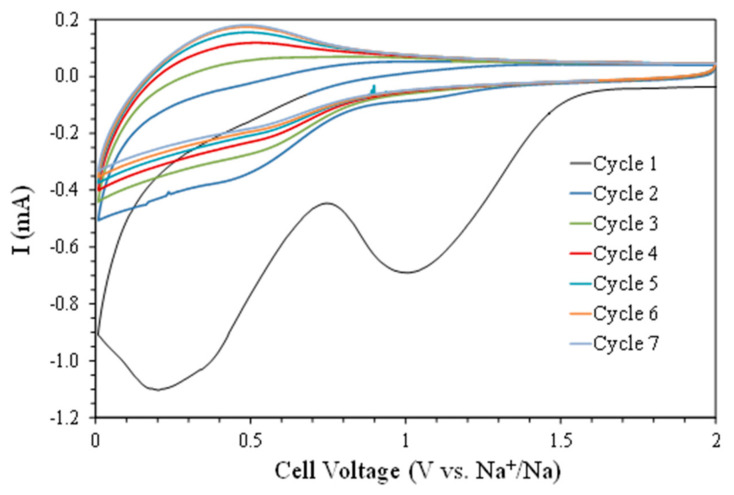
Cyclic voltammetry (CV) of a cell with a 3D rGO electrode for the first seven cycles between 0.010 and 2.000 V versus Na^+^/Na at 6.0 mV min^−1^.

**Figure 4 materials-16-05386-f004:**
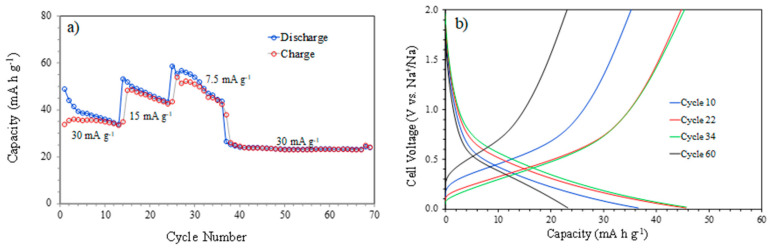
Galvanostatic discharge–charge test of a graphene sample between 0.010 and 2.000 V vs. Na^+^/Na: (**a**) discharge and charge capacities versus cycle number at different current densities and (**b**) capacities versus cell voltage for specific cycles.

**Figure 5 materials-16-05386-f005:**
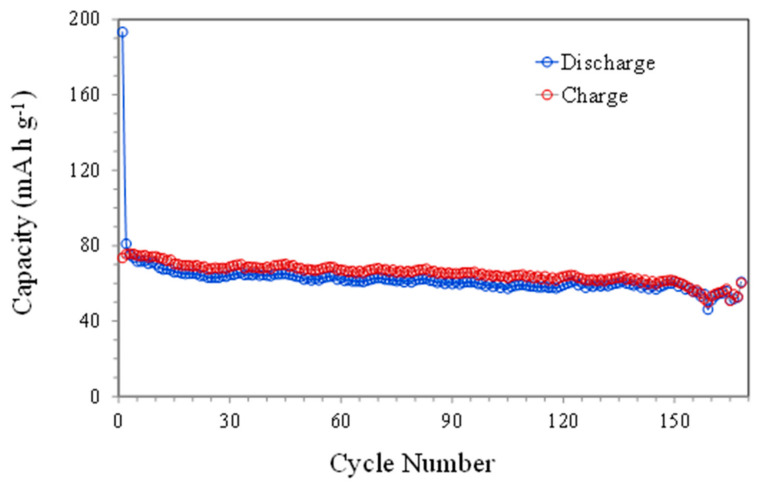
Discharge and charge capacities versus cycle number at 10 mA g^−1^.

**Figure 6 materials-16-05386-f006:**
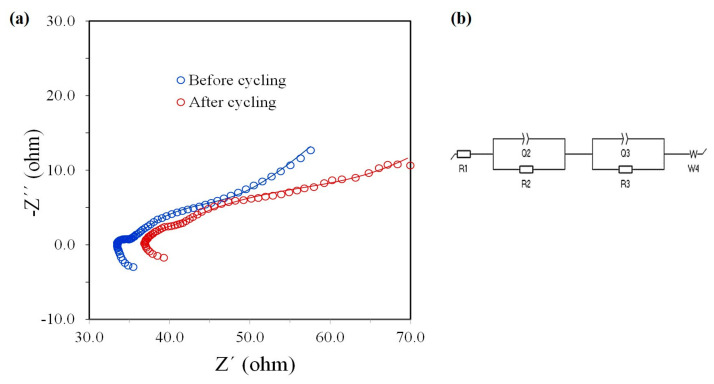
(**a**) Nyquist plots before and after charge–discharge cycling. (**b**) Equivalent circuit model for curve fitting (continuous lines in (**a**)).

**Figure 7 materials-16-05386-f007:**
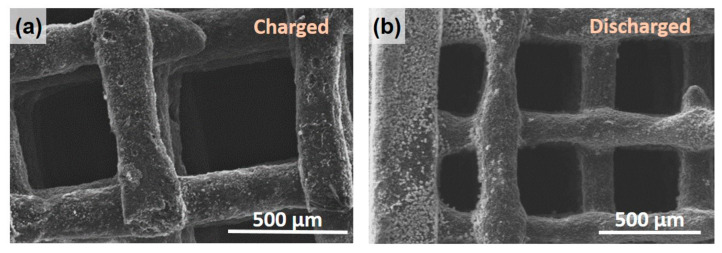
SEM images of (**a**) 3D rGO charged and (**b**) discharged electrodes at different magnifications: the reacted material covered the surfaces of the structures inhomogeneously.

**Figure 8 materials-16-05386-f008:**
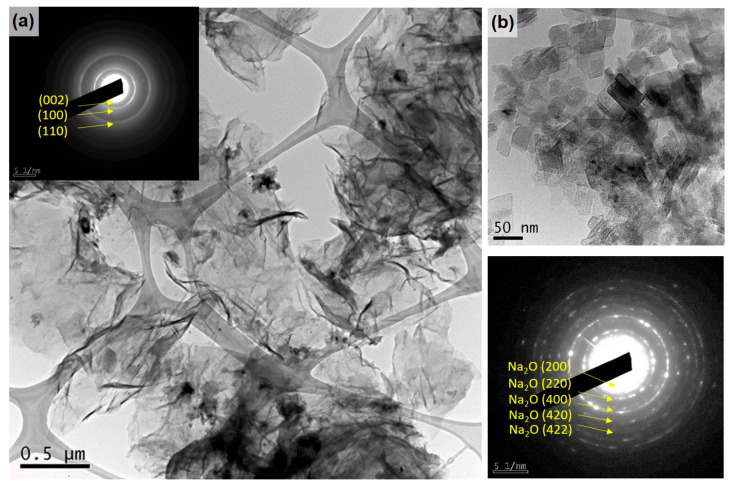
(**a**) TEM image of a discharged rGO electrode with inset showing the SAED pattern of the rGO. (**b**) Higher-magnification image of aggregated nanosized cubic-like crystals and corresponding SAED with identification of the observed reflections of Na_2_O.

**Figure 9 materials-16-05386-f009:**
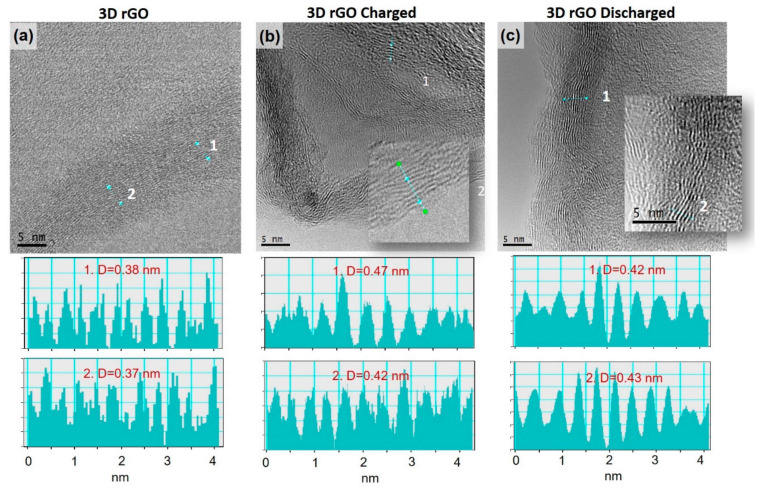
HRTEM images of graphene sheets in original rGO (**a**), charged (**b**), and discharged (**c**) electrodes with examples of representative interplanar spaces.

**Figure 10 materials-16-05386-f010:**
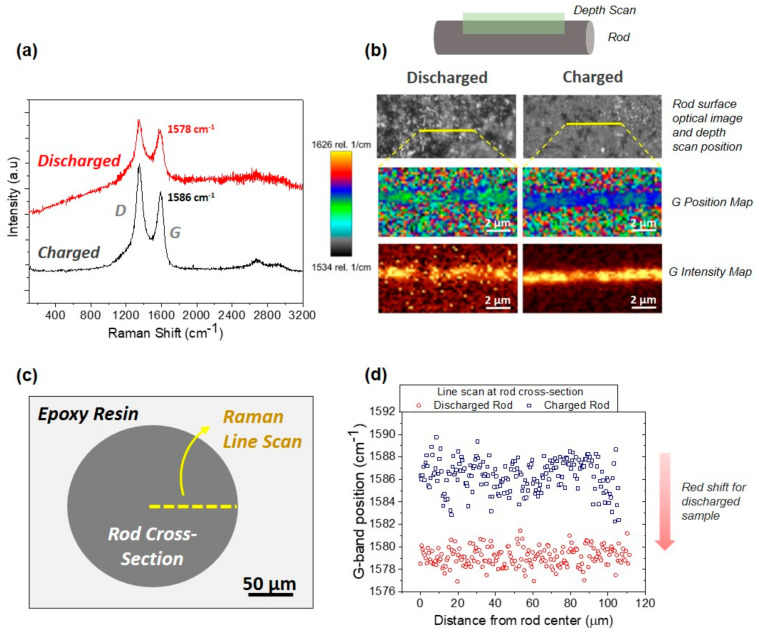
Raman results for the charged and discharged electrodes taken along the filament surface region (up to 3 µm in depth) and radially through the filament cross-sections. (**a**) Characteristic Raman spectra averaged over surface measurement. (**b**) Optical images of both samples showing the location of the depth scan, corresponding G-band position maps with a scale range of (1534–1626 cm^−1^), and G intensity maps along and below these lines. (**c**) Schematic diagram of measurement of the rod cross-section showing the position of the line scan, whose results are shown in d. (**d**) G-band position versus distance from rod center.

**Table 1 materials-16-05386-t001:** Density and porosity of the 3D rGO samples.

ρ_bulk_ (g · cm^−3^)	ρ_solid_ (g · cm^−3^)	P_skeleton_ (%)	P_macro_ (%)	P_total_ (%)
0.13	0.38	83	62	94

**Table 2 materials-16-05386-t002:** Characteristic bands of Raman spectra for the 3D rGO structures.

Sample *	D (cm^−1^)	G (cm^−1^)	I_D_/I_G_
3D rGO Na/charged, in depth	1350 ±1	1586 ±1	1.9 ±0.1
3D rGO Na/discharged, in depth	1350 +/−1	1578 +/−1	1.7 +/−0.1
3D rGO original	1355 ±1	1596 ±1	1.9 ±0.1

* Polished samples.

**Table 3 materials-16-05386-t003:** Reported G-band position redshift measured during sodiation for electrodes fabricated with different carbon materials.

Electrode Material	Excitation Wavelength (nm)	Δ G-Band Position (cm^−1^) by Insertion/De-Insertion	Reference
Hard carbon synthesized from coconut shell	633	50	[42]
Commercial hard carbon	----	25	[43]
Commercial plant-based hard carbon	532	38	[44]
Ground pyrolyzed carbon	532	38	[47]
	633	41	
	780	42	

**Table 4 materials-16-05386-t004:** Examples of capacities and cycling stability for rGO-based electrodes obtained using conventional cast methods, aerogel routes, or 3D-printing techniques.

Electrode Material	Design	Cell Test	Discharge Capacity (mAh g^−1^)	Cycle Life	Reference
rGO/carbon black/PVDF	Bulk, 100 µm thickness	Coin cell	141 at 40mA g^−1^	1000 at 40 mA g^−1^	[31]
rGO/carbon black/PVDF	Bulk, 30 µm thickness	Coin cell	235 at 30 mA g^−1^	1000 at 30 mA g^−1^	[18]
rGO/carbon black/methyl cellulose	Bulk	Coin cell	603 at 0.05 A g^−1^	10000 at 5 A g^−1^	[15]
Holey rGO/carbon black/PTFE	Bulk	Coin cell	365 at 0.1 A g^−1^	3000 at 2 A g^−1^	[49]
rGO/CNT paper	Bulk, 12 µm	Coin cell	166 at 0.05 A g^−1^	300 at 200 mA g^−1^	[50]
rGO	Aerogel	Coin cell	250 at 0.05 C	20 at 0.05 C	[14]
Cyclodextrin rGO	Aerogel	Coin cell	500 at 0.05 C	100 at 1 C	[14]
rGO/carbon black/PVDF	Foam	Coin cell	800 at 0.1 A g^−1^	150 at 500 mA g^−1^	[17]
Na@rGO	3D, 250 µm thickness	Coin cell		500 at 1 mA cm^−2^	[21]
rGO/Na_3_V_2_(PO_4_)_3_	3D-printed	Coin cell (full cell)	95 at 0.1 A g^−1^	1000 at 100 mA g^−1^	[22]
Na@rGO/CNT	3D-printed	Coin cell		640 at 8 mA cm^−2^	[23]
rGO/Na_3_V_2_(PO_4_)_3_	3D-printed	Coin cell	1.26 mAh cm^−2^ (areal capacity) at 0.2 C	900 at 1 C	[24]
rGO (Na-O_2_ battery)	3D-printed	Swagelok cell	500 at 0.5 A g^−1^	122 at 0.5 A g^−1^	[25]
rGO	3D-printed	Three-electrode cell	80 at 10 mA g^−1^	150 at 10 mA g^−1^	This work

## Data Availability

All data are contained within the article and Appendix A.

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
