# Peer review of "Electrochemical Response of 3D-Printed Free-Standing Reduced Graphene Oxide Electrode for Sodium Ion Batteries Using a Three-Electrode Glass Cell"

_materials, 2023, doi:10.3390/ma16155386_

Round 1

Reviewer 1 Report

-          The abstract and conclusion should be rewritten to contain the most interesting results and estimated values.

-          The introduction part should be updated with recent and relevant publications about the porous carbon-based anode materials such as Diamond and Related Materials 121 (2022) 108722, Journal of Molecular Structure 1251(2022) 131964.

-          XPS of C 1s and O 1s is recommended to be measured.

-          Cyclic voltammetry studies are recommended to confirm and explain the electrochemical storage mechanism.

-          Electrochemical Impedance Spectroscopy (EIS) of the half charged cells should be investigated.

-          The obtained results of Charge-discharge capacity, Cycle life and EIS parameters for 3D printed rGO should be compared with bulk rGO anode materials after cycling versus Na/Na+ electrode.

There are many typing and grammatical errors that need to be checked carefully.

Author Response

  1. The abstract and conclusion should be rewritten to contain the most interesting results and estimated values.

We thank the reviewer´s comment. Both sections have been revised and now include the capacity achieved and principal observations on electrolyte degradation as well as on Na insertion.

  1. The introduction part should be updated with recent and relevant publications about the porous carbon-based anode materials such as Diamond and Related Materials 121 (2022) 108722, Journal of Molecular Structure 1251(2022) 131964.

We thank the reviewer for the recommendation and also we carefully checked the cited references. Unfortunately, we can´t follow the recommendation of including between the references the paper The influence of the drying method on the microstructure and the compression behavior of graphene aerogel by Jing Xie et al., Diamond & Related Materials 121 (2022) 108772.  This paper compares two methods of drying GO aerogels (supercritical and freeze drying) on the porous characteristics and robustness of the aerogel. It doesn´t have any relation with the electrochemical behavior of rGO that is the aim of present research. As for the paper Effect of polymerization conditions on the physicochemical and electrochemical properties of SnO2 /polypyrrole composites for supercapacitor applications by Fahim Hamidouche et al., Journal of Molecular Structure 1251 (2022) 131964, we do not see either any relation with the subject of present manuscript, hence we find difficult to include it in our introduction.

Nonetheless, following the recommendation of including relevant and recent publications, we have updated the introduction with additional contributions: ref 4, Red Phosphorus nanodots on Reduced Graphene Oxide as flexible and ultrafast anode for sodium-ion batteries by Yihang Liu et. al. ACS Nano 11 (2017) 5530-5537, proposed by reviewer 3, and ref. 26, 3D printed Au/rGO microlattice host for dendrite-free sodium metal anode by Hui Wang Energy storage materials 55 (2023) 631-641.

  1. XPS of C 1s and O 1s is recommended to be measured.

The authors are again grateful to the reviewer for the suggestion but we consider that the XPS measurements, though it is a useful method to investigate electrode-electrolyte interphase, are beyond the scope of the present work.  Nevertheless, the information on the amount of O2 and C in rGO, which has been already reported in ref. 27 is also mentioned in page 5 giving a rGO oxygen content of 1wt.% after thermal treatment.

  1. Cyclic voltammetry studies are recommended to confirm and explain the electrochemical storage mechanism.

Cyclic voltammetry is studied in the manuscript and consequently described in section 2.2 Electrochemical measurements, it was carried out in the range 0.010-2.000 V vs Na+/Na at 6 mV/min. Reviewer can find these results in pages 5 and 6, and Fig. 3.

  1. Electrochemical Impedance Spectroscopy (EIS) of the half-charged cells should be investigated.

We thank the reviewer for this suggestion and a new part showing the results of EIS measurements has been included in the manuscript. The experimental set up is described in section 2.2 and results and discussion were added to pages 7 and 8 (also see new Figure 6).

  1. The obtained results of Charge-discharge capacity, Cycle life and EIS parameters for 3D printed rGO should be compared with bulk rGO anode materials after cycling versus Na/Na+ electrode.

This recommendation will be taken into account in a future work on studying the effect of strut density, 3D printed design and rGO reduction level to the improvement of specific capacity and stability of free-standing electrode. As shown in the introduction, the evaluation of rGO 3D printed structures in Na battery applications has been carried out for improving current density distribution in combination with Na metal anode or forming composite structures with active materials. For this reason, the aim was to observe electrochemical behavior of pure rGO 3D printed structure. To improve information regarding bulk electrodes, reported data mainly obtained using the coin cell configuration have been explicitly included in new table 4. However, we would like to asses that the direct comparison with those data is not straightforward due to the lack of information or the differences on testing conditions in some of the works.

Reviewer 2 Report

The authors work on 3D printed rGO electrodes for Na-ion batteries is interesting, very well written, and demonstrates a broad and detailed investigation. 

It is very nearly suited for publication, as it is. However, as I read this work I find no mention anywhere of the conventional type of Na-ion battery electrode - a thin film coating on a metallic foil. So from this view, I need to see what is the point of going into exploration of 3D printed electrodes that are not even compressed. 

The paper needs a stronger description of why this unconventional approach to making electrodes is practical important / necessary energy storage research, or else clearer communication of how it is curiosity driven research that has led to new insights about sodium ion (de)insertion in carbon-based electrodes.  

Please edit that abstract to clarify the possible practical application/s for these electrodes, or alternatively, emphasize the new insight revealed from an interesting lab study.

It is recommended that the authors also use the revision as an opportunity to strengthen their work by editing some unproven hypotheses that could be easily verified. E.g. the electrolyte turns yellow indicates degradation of the solvent.

Line 205: check whether "punctual" is the best word

Line 297: perhaps "no" is meant to be "not"?

Author Response

  1. As I read this work I find no mention anywhere of the conventional type of Na-ion battery electrode - a thin film coating on a metallic foil. So from this view, I need to see what is the point of going into exploration of 3D printed electrodes that are not even compressed. 

The studies on 3D printed batteries have increased in recent years as additive manufacturing is marking a new paradigm in low-cost production and waste reduction. Authors consider relevant the observation of structure behavior as-printed (not compressed) to increase information on whether free-standing pure rGO structure would achieve good cycling stability, what is the range of capacities obtained and the suitability for sodium insertion. From this point on it would allow to continue investigating on different designs, materials treatment and electrolyte selection for improving this behavior.

  1. The paper needs a stronger description of why this unconventional approach to making electrodes is practical important / necessary energy storage research, or else clearer communication of how it is curiosity driven research that has led to new insights about sodium ion (de)insertion in carbon-based electrodes.  

Additive manufacturing techniques, as mentioned in the previous point, have brought new possibilities in battery fabrication that are beyond the properties obtained by using conventional bulk electrodes. These advantages are the flexibility in shape and suitability for designs with multiscale porosity, which are limited for components produced by traditional casting techniques.  Examples of the unique configurations obtained are the interdigitated, coaxial, lattice, twisted fibers or non-planar structures which infill pattern can be also used to infiltrate polymer, gel or liquid electrolytes. Thus, research on rGO lattice structures contribute to increase information on the use of graphene based materials in energy storage, the sodium insertion in pure rGO material with the utilized density and conductivity or particular challenges that need to be overcome when using 3D printed structures.

A new paragraph has been added to introduction highlighting the importance of thick designs.

  1. Please edit that abstract to clarify the possible practical application/s for these electrodes, or alternatively, emphasize the new insight revealed from an interesting lab study.

The abstract has been modified to emphasize importance of this type of electrodes.

  1. It is recommended that the authors also use the revision as an opportunity to strengthen their work by editing some unproven hypotheses that could be easily verified. E.g. the electrolyte turns yellow indicates degradation of the solvent.

The authors thank the reviewer comment. A new sentence has been added in the form of hypothesis to the end of introduction section. In this sense the result and discussion show that degradation products and yellow color in electrolyte system have been also observed using NaClO4-PC and hard carbon materials. A new reference (Journal of Electrochemical Chemistry 895 (2021) 115505) Cycling degradation and safety issues in sodium-ion batteries, promises of electrolyte additives has been added regarding the yellow color.

  1. Line 205: check whether "punctual" is the best word

The word “punctual” has been substituted by “occasional”

  1. Line 297: perhaps "no" is meant to be "not"?

Misspelling has been corrected

Reviewer 3 Report

Throughout this research work, good scientific achievements are provided that are in the development of 3D printed, free-standing reduced graphene oxide electrode for next generation sodium ion batteries applications. This work is very interesting and provides lots of scientific knowledge about Na-ion battery technology for post-lithium battery technology. The manuscript is perfectly match for the publication in this journal. However, a major revision is required as indicated below in order to eliminate some shortfall of this work:

  1. Language and formats need to be thoroughly checked to avoid any mistakes before acceptance.
  2. Authors are suggested to provide electrochemical impedance spectroscopy (EIS) measurement before and after cyclic test with suitable equivalent circuit model for better understanding.
  3. A comparison table of reduced graphene oxide electrode for sodium ion batteries performance can be provided to improve the quality of this work.
  4. These  publications could be informative for authors and are suggested to cite in proper place . Scientific Reports volume 7, 40910 (2017); Materials 2021, 14, 2942; ACS Nano 2017, 11, 5530−5537.

Language and formats need to be thoroughly checked to avoid any mistakes before acceptance.

Author Response

1. Language and formats need to be thoroughly checked to avoid any mistakes before acceptance.

We thank the reviewer for the comment. Language has been revised and grammatical errors have been corrected to improve readability.

  1. Authors are suggested to provide electrochemical impedance spectroscopy (EIS) measurement before and after cyclic test with suitable equivalent circuit model for better understanding.

Following reviewer suggestion as well as that of reviewer 1 (see point 5 of R1 comments) EIS results with the equivalent circuit model have been included in the manuscript.

  1. A comparison table of reduced graphene oxide electrode for sodium ion batteries performance can be provided to improve the quality of this work.

Special attention was put in the introduction section to reference papers related to 3D printed rGO structures applied as components in Na batteries (refs. 14 to 25). The aim of these works have been the improvement of current density through rGO current collector or hybrid rGO/Na metal anode and the development of pure rGO electrode for Na-O2 battery. A comparison table (Table 4) gathering these results and other examples of bulk anodes has been included in the manuscript in order to provide a more complete description.

  1. These publications could be informative for authors and are suggested to cite in proper place. Scientific Reports volume 7, 40910 (2017); Materials 2021, 14, 2942; ACS Nano 2017, 11, 5530−5537.

We thank the reviewer for suggesting these references. The Materials 2021, 14, 2942 publication has been added to discussion´s final paragraph as example of alternative electrolyte. Scientific Reports volume 7, 40910 (2017) was already cited in the Introduction and it has been also included in new Table 4 Examples of capacities and cycling stability for rGO based electrodes obtained by different methods as example of thin bulk electrode.  Also, ACS Nano 2017, 11, 5530−5537 publication has been added to the introduction (ref. 4) as a case of hybrid material that enhances the performance of rGO electrode.

Round 2

Reviewer 3 Report

The revised manuscript is suitable for acceptance in this journal. Editors can make their final decision.